# Alternative Mechanisms of p53 Action During the Unfolded Protein Response

**DOI:** 10.3390/cancers12020401

**Published:** 2020-02-10

**Authors:** Leïla T. S. Fusée, Mónica Marín, Robin Fåhraeus, Ignacio López

**Affiliations:** 1INSERM U1162, 27 rue Juliette Dodu, 75010 Paris, France; 2Biochemistry-Molecular Biology, Faculty of Science, Universidad de la República, Iguá 4225, 11400 Montevideo, Uruguay; 3RECAMO, Masaryk Memorial Cancer Institute, Zluty kopec 7, 656 53 Brno, Czech Republic; 4Department of Medical Biosciences, Umeå University, 90185 Umeå, Sweden; 5ICCVS, University of Gdańsk, Science, ul. Wita Stwosza 63, 80-308 Gdańsk, Poland

**Keywords:** p53, p47, ER stress, UPR, mRNA translation

## Abstract

The tumor suppressor protein p53 orchestrates cellular responses to a vast number of stresses, with DNA damage and oncogenic activation being some of the best described. The capacity of p53 to control cellular events such as cell cycle progression, DNA repair, and apoptosis, to mention some, has been mostly linked to its role as a transcription factor. However, how p53 integrates different signaling cascades to promote a particular pathway remains an open question. One way to broaden its capacity to respond to different stimuli is by the expression of isoforms that can modulate the activities of the full-length protein. One of these isoforms is p47 (p53/47, Δ40p53, p53ΔN40), an alternative translation initiation variant whose expression is specifically induced by the PERK kinase during the Unfolded Protein Response (UPR) following Endoplasmic Reticulum stress. Despite the increasing knowledge on the p53 pathway, its activity when the translation machinery is globally suppressed during the UPR remains poorly understood. Here, we focus on the expression of p47 and we propose that the alternative initiation of p53 mRNA translation offers a unique condition-dependent mechanism to differentiate p53 activity to control cell homeostasis during the UPR. We also discuss how the manipulation of these processes may influence cancer cell physiology in light of therapeutic approaches.

## 1. Introduction

The p53 tumor suppressor is mainly known as a transcription factor that both positively and negatively regulates the expression of a diverse multitude of genes following different insults, such as DNA damage, nutrient deprivation, viral infection, and oncogene activation, among others [1]. Due to its central role in cellular homeostasis and physiological processes, *TP53* is the most frequently mutated gene in human cancers, as recently confirmed by the analyses of the Catalogue Of Somatic Mutations In Cancer (COSMIC) [2] and The Cancer Genome Atlas (TCGA) Pan-Cancer effort [3]. Most of the mutations identified are located in p53’s DNA-binding (DB) domain and result in a transactivation-deficient protein [2,4].

Besides somatic alterations, germline mutations in the human *TP53* gene constitutes an enhanced risk of developing a wide spectrum of early-onset cancers, as they are one of the underlying causes of a rare familial cancer disorder called Li-Fraumeni syndrome [5,6]. The cancers most often associated with this syndrome include breast cancer, osteosarcoma, soft-tissue sarcomas, brain tumors, adrenocortical carcinomas, and leukemia, particularly in children and young adults [6]. Patients with this syndrome generally express both the mutant and wild-type (p53wt) forms of p53 in all tissues. During cancer progression, the wild-type activity of the protein is often lost, either due to the occurrence of dominant-negative (DNE) inhibitor mutations, to a gain of function (GOF) mutation that favors cancer progression, or to a direct loss of p53wt allele, a phenomenon known as loss of heterozygosity (LOH) [1,6]. The important handicap imposed by expressing half of the normal amount of fully active p53 in Li-Fraumeni patients [1,7] highlights the sensitivity of the pathway to small changes in p53 levels.

In tumor cells containing wild-type *TP53* gene, p53 activity might be compromised through different mechanisms. A well-known example constitutes the inhibitory interaction of p53 with proteins from cancer-associated virus, such as the T antigen from SV40 [8,9], adenovirus E1b protein [10] and the E6 protein from human papilloma virus (HPV) types 16 and 18 [11,12]. Overexpression of cellular regulators such as Mouse double minute 2 homolog MDM2 [13] and its homolog MDMX (MDM4) [14] can also suppress p53 activity and therefore have oncogenic potential. Under normal conditions, MDM2 and MDMX bind the conserved BOX-I motif in the N-terminus of p53 and mask its transactivation (TA) domain [13,14,15,16]. Moreover, MDM2, but not MDMX, possesses an E3-ubiquitin ligase activity that relies on its C-terminal RING domain, and targets p53 for 26S-dependent proteasomal degradation [17].

p53 activation during the DNA damage response (DDR) has been well studied and includes a direct and indirect phosphorylation by the ATM kinase that prevents the interaction with MDM2 and induces its transcription activity [18,19]. Once activated, p53 stimulates and suppresses different sets of gene products that aim to either prevent abnormal growth by a reversible arrest of the cell cycle to facilitate repair processes, or to induce irreversible outcomes including apoptosis or senescence [20,21,22,23,24]. Two of the best-described p53 target genes are *p21^Cip1/Waf1^* (hereafter p21) and *mdm2* itself [16,20,25,26]. Induction of p21 in early stages of the DDR suppresses both G1 and S phase cyclins and cyclin-dependent kinases (CDKs), and therefore prolong the G1 phase to allow the cells to repair the damage before DNA replication occurs [20,25]. Induction of MDM2 and the ATM-mediated phosphorylation of MDM2 and MDMX, however, constitute a positive regulatory loop towards p53 activation that includes an increase in its half-life and in the rate of p53 protein synthesis [27,28]. The latter depends on MDM2 and MDMX’s capacity to bind p53’s mRNA through their C-terminal RING domains [27,28,29].

The above-described pattern of p21 and MDM2 expression is not observed upon Endoplasmic Reticulum (ER) stress. Indeed, in cultured cells facing ER stress, expression of both proteins is down-regulated in a post-transcriptional and p53-dependent manner that relies on regulatory elements located in their respective mRNA coding sequences [30,31]. This points towards a shift in p53 activity during the Unfolded Protein Response (UPR) that is, at least in part, mediated at the level of mRNA translation. The following sections describe this activity of p53 and the associated physiological consequences.

## 2. Stress to the Endoplasmic Reticulum Triggers the Unfolded Protein Response

The ER is the main subcellular compartment involved in protein folding and maturation, where around one-third of the total cell proteome is processed. When the protein folding capacity of the ER is insufficient and an imbalance between cellular demand and ER function threatens the cellular homeostasis, the cells activate the UPR. There are several physiological and pathological conditions that can trigger ER stress, namely: glucose starvation, underglycosylation of glycoproteins, modification of calcium flux across the ER membrane, elevated protein synthesis and secretion, failure of protein folding, transport or degradation, oxidative stress, among others [32,33,34]. These are all hallmarks of various disease-related processes and thus, ER stress and the activation of the UPR have attracted much interest from academia as well as from the industry.

The UPR is classically described as an adaptive three-branched pathway that aims to restore the balance between newly synthesized and mature proteins and has been extensively reviewed within this special issue and elsewhere [33,35,36,37]. The three branches of the canonical UPR pathway are represented by three transmembrane proximal sensors: inositol-requiring enzyme 1α (IRE1α) and IRE1β, protein kinase RNA-like ER kinase (PERK), and activating transcription factor 6 (both α and β isoforms and referred to as ATF6) [35,36,37]. In normal conditions, their activity is blocked by binding of their luminal domains to the heat-shock ER resident chaperone Glucose-regulated protein 78/Binding immunoglobulin protein GRP78/BiP and other associated factors [36,37]. Triggering of the UPR releases the ER stress sensors from BiP, leading to their activation and recovery of protein folding capacity [36,38]. This includes a general down-regulation of protein synthesis and stimulation of ER-associated degradation of misfolded proteins (ERAD). It is not surprising that the UPR is exploited by cancer cells to handle stress induced by growth-limiting conditions encountered in the tumor environment, such as hypoxia, nutrient deprivation, acidosis, redox imbalance, exposure to drugs, etc. Ultimately, adapting to such conditions results in sustained cancer survival and chemoresistance [37,39,40,41,42].

However, if cells are unable to cope with the stress, the UPR engages the cell into a pro-death signaling pathway that leads to mitochondria-dependent apoptosis [34,35,43]. This phenomenon is thought to explain the anti-cancer effects of proteasome inhibitors (PIs) that induce accumulation and aggregation of proteins that are less tolerated by the highly proliferative and protein synthesis-demanding malignant cells (see below) [44,45]. Yet, the molecular switches mediating the transition from adaptive to pro-apoptotic responses are not fully understood. Nevertheless, some important actors, among which is p53, have been proposed to promote cell death during the UPR [34,43,46].

## 3. Expression and Activity of p53 During ER Stress

### 3.1. Activation of p53 During ER Stress

Albeit suggested as an essential player, the role of p53 during the UPR is still not fully understood. Stabilization of p53 was reported upon treatment of HCT116 and U2OS cell lines as well as primary mouse embryonic fibroblasts (MEFs) with tunicamycin or glucose deprivation in a PERK-dependent manner [47]. This was suggested to be due to a stronger association between MDM2 and ribosomal proteins RPL5, RPL11, and RPL23, leading to reduced MDM2-mediated p53 ubiquitination and degradation. This, in turn, is caused by the reduction of active polysomes after eIF2α phosphorylation [47]. Activation of p53 after treatment with tunicamycin, thapsigargin or brefeldin A, was also reported in primary human trabecular meshwork cells, Mel-RM and MM200 melanoma, and in breast MCF-7 and cervical HeLa cancer cell lines [48,49,50]. Further, Li et al. have shown that p53 nuclear localization and activity were increased in MEFs incubated with thapsigargin and tunicamycin [46]. Together, these observations suggest that ER stress-mediated induction of p53 appears as a common response in different types of cells exposed to diverse ER stress chemical inducers.

Activation of p53 during ER stress was mainly associated with increased apoptotic cell death acting through different pathways. Indeed, activation of nuclear factor kappa-light-chain-enhancer of activated B cells (NF-κΒ), and p53 upregulated modulator of apoptosis (PUMA) and NOXA, two well-described pro-apoptotic B-cell lymphoma 2 (Bcl-2) proteins, was observed together with p53 in MCF-7, HeLa and in MEF cells, respectively [46,50]. However, ER stress-induced apoptosis was only partially suppressed in *p53^-/-^* MEFs, suggesting that other pathways could be involved [46]. For instance, induction of the ER-linked pro-apoptotic C/EBP homology protein CHOP (also named growth arrest and DNA-damage-inducible 153, GADD153), which promotes apoptosis by favoring translation recovery of cell-death proteins [51], remained intact in p53-negative MEFs [46]. This shows that both p53-dependent and -independent pathways may trigger apoptosis upon ER stress [34,43,46], the latter becoming more important in those cases where p53 is down-regulated.

### 3.2. Inhibition of p53 During ER Stress

It has been also observed that p53 adopts a mostly cytoplasmic distribution during ER stress. This phenomenon was reported to be a response to phosphorylation by the ER stress-dependent PERK and protein kinase RNA-activated (PKR)-induced glycogen synthase-3 (GSK-3) kinase in human diploid WI-38 and human fibrosarcoma HT1080 cells incubated with either thapsigargin, tunicamycin or glucose-free medium [52,53]. The cytoplasmic localization correlated with higher ER stress-induced and MDM2-associated degradation of p53, following its phosphorylation by GSK-3 that targeted it for proteasomal processing. Significantly, the cooperative action of GSK-3 and MDM2 also occurred in unstressed cells, but it appeared enhanced in cells subjected to ER stress [54]. MDM2-dependent ubiquitination and concomitant degradation of p53 was also observed during the early phase of thapsigargin-induced ER stress response in cultured cortical neurons [55]. Additionally, Synoviolin (also called HRD1) an E3-ubiquitin ligase proposed to be part of the ERAD complex [56], was also reported to sequester p53 to a perinuclear/ER region and target it for degradation in the cytoplasm [57]. Indeed, down-regulation of Synoviolin by siRNA in the p53wt human colon cancer cell line RKO, resulted in increased p53 protein level and nuclear accumulation as well as in triggering of the UPR [57].

Synthesis of p53 protein was also shown to be affected in p53-proficient and p53-overexpressing established cell lines by the general inhibition of cap-dependent translation during thapsigargin or tunicamycin-promoted ER stress. This effect was accomplished via PERK phosphorylation of eIF2α and showed that general inhibition of cap-dependent mRNA translation contributes to dampen p53 expression (Figure 1) [58,59]. On the contrary, it was observed that the expression of the shorter p47 isoform was specifically promoted under such circumstances.

### 3.3. Induction of The p47 Isoform During ER Stress

The above-presented data on the role of p53 during the UPR focus on the full-length p53 (p53fl) protein without addressing the expression of its isoforms. So far, 12 isoforms have been described as the combinatorial result of 4 N-terminal and 3 C-terminal variants [64]. The former is referred to as Δ and are incremental deletions of the amino-terminal portion of the protein, therefore the Δ40, Δ133, and Δ160 variants lack the first 39, 132 and 159 residues of the canonical isoform, respectively. Alternative splicing of intron 9 gives rise to 3 C-terminal variations named α, β and γ. The former represents the canonical protein while the other 2 isoforms are shorter versions with different C-terminal regions that lack part of the oligomerization domain and the entire regulatory C-terminal domain of the p53fl canonical protein (p53α) (Figure 1) [64]. The mechanisms by which these isoforms arise are beyond the scope of this article and have been extensively reviewed elsewhere [64,65,66]. In this review, we will focus on the roles of p53fl and Δ40p53α (also called p53ΔN40, p53/47 and referred to here as p47) (Figure 1) during ER stress.

Translation of p47 from the second AUG in the *p53* mRNA is specifically promoted when cap-dependent translation is compromised in response to ER stress [58,59,67,68]. The +1 to +118 coding region of *p53fl* mRNA corresponds to the 5’ UTR of *p47* and contains an Internal Ribosome Entry Site (IRES) [59,61,69]. Several IRES-transacting factors (ITAFs) were shown to bind to this region and tune the expression of both p53fl and p47 [66]. Moreover, other numerous proteins apart from ITAFs were shown to interact throughout the *p53* mRNA in response to different signaling pathways and control the RNA stability, translation efficiency or alternative initiation of translation, emphasizing the seminal role of *p53* mRNA as a platform that orchestrates the p53 pathway [70,71,72].

The signal transducer between the ER status and p47 expression is the PERK kinase (Figure 1). Indeed, co-expression of the dominant-negative PERKΔC mutant or suppression of PERK using small interfering RNA, inhibited thapsigargin-induced p47 translation [58]. Upon ER stress, increased expression of p47 leads to G2 arrest that serves two purposes. First, cap-independent translation is the preferred mechanism of translation during G2 and by reducing general cap-dependent protein synthesis, p47 contributes to restore the balance between protein synthesis and protein folding in the ER [73,74]. Secondly, it favors the cap-independent translation of a selected number of mRNAs, such as *ATF4*, *cat-1* and *p53*, whose products take part in damage control and/or promote cell death in advanced non-repairable stages [75,76,77]. Overall, this can be compared with the G1 arrest caused by p53fl following DNA damage to allow DNA repair [20,25].

## 4. The Short p47 Isoform

### 4.1. Roles of p47

The short p47 isoform lacks the first 39 aa of p53fl, including the transactivation domain I (TAI) and the binding site for several proteins, most notably that of MDM2 and MDMX (Figure 1). As a consequence, p47 has a prolonged half-life and different activity compared to p53fl [62,78]. Furthermore, it retains the oligomerization and DNA-binding domains and can affect p53 activity by establishing homo- or hetero-oligomers, therefore controlling a different stress-dependent set of target genes [63,79,80].

The functional properties of p47 have been addressed in vitro and in animal models by several groups. It has been observed that induction of pro-apoptotic BAX protein was similar in *p53*-null H1299 cells overexpressing either p53fl or p47 isoforms [62]. More recent reports showed in addition that p47 retains the capacity to induce some specific apoptosis-related genes such as *PIG3* and *AIP1*, among others not induced by p53fl, such as *P53BP2* or *TIAL1* [63,81]. This is supported by the fact that both p53fl and p47 were able to induce apoptosis with similar efficiency, as evidenced by the sub-G0 population estimated by fluorescence-activated cell sorting (FACS) and by their similar capacity to inhibit cell growth in colony formation assays (CFA) under normal conditions [62,81]. Conversely, other groups have shown that overexpression of p47 did not suppress the growth of *p53*-null cell lines in clonogenic assays but on the contrary, it counteracted the suppressive effect of p53fl [60,82]. In line with this observation, p47 was not able to activate the transcription of reporter constructs containing responsive elements from different p53 target genes (*PIG3*, *GADD45*, *Cyclin-g*, *MDM2,* and *p21*) to the same extent as p53fl [60,62,63,82]. Altogether, these data thus point towards different and specific activities of p47 that rely on its transactivation II (TAII) domain, which would promote particular responses of p53-regulated genes depending on the cellular context. Indeed, these seemingly contradictory results might implicate that p47 acts specifically under different conditions and/or that the ratio of p47:p53fl might be determinant to promote an adequate response.

### 4.2. Roles of p47 in vivo

The establishment of hetero-oligomers of p53fl with its isoforms can prevent MDM2-induced degradation of p53fl [60,62,82] and can alter its activity as a transcription factor. Indeed, mice heterozygous for a deletion mutation encompassing the first six exons of *p53* and encoding only a C-terminal p53 fragment showed an early onset of phenotypes associated with aging, including reduced longevity, osteoporosis, generalized organ atrophy and diminished stress tolerance [83]. A similar phenotype, together with resistance to spontaneous tumor formation was observed in transgenic mice overexpressing p44 (mouse homolog of p47). Cells from p44-overexpressing animals showed an abnormal insulin-like growth factor (IGF) signaling pathway that was proposed to be responsible for the growth suppression and senescence, and for the observed premature aging phenotype (Figure 2) [84]. Importantly, this phenotype was reversed when p44-overexpressing mice were crossed with *p53*-null animals [83,84], supporting the notion that p44 requires the presence of p53fl to exert its activity.

It is likely that the activity of p47 also depends on the physiological state of the cell. For example, the aging phenotype was shown to depend on the level of reactive oxygen species (ROS), whose high reactivity towards DNA, proteins and membrane lipids would promote ER stress, among others [32,33,85]. Furthermore, a gene expression signature linked to p53 and oxidative stress that promotes cell cycle arrest, senescence and apoptosis was reported throughout the aging process [86]. This response pathway induces G2 arrest and was active only in the context of wild-type p66Shc, a key factor in controlling intracellular ROS levels [85]. Interestingly, in p66Src-negative MEFs, expression of p44 was impaired to a higher extent than that of p53fl, suggesting that the effect on transcription and the subsequent arrest of cells at the G2/M phase of the cycle is largely dependent on p44 [86].

The specific role of p47 in a particular tissue is indicated by the observation that neurons from mice displaying the aging phenotype described above showed an atypical high phosphorylation of Tau microtubule-binding protein. Hyper-phosphorylation of Tau was carried out by p44-induced and p53fl-independent Dyrk1A, Gsk3beta, Cdk5/p35/p39 Tau kinases (Figure 2). This led to destabilization of microtubules which in turn promoted synaptic deficits and cognitive decline, all hallmarks of aged brains and Alzheimer’s disease [87]. Up-regulation of p44 was proposed to be dependent, at least in part, on increased expression and processing of amyloid precursor protein (APP) during aging [88], whose intracellular domain (AICD) was shown to bind *p53*’s second IRES and promote cap-independent translation of a luciferase reporter gene [89].

Intriguingly, p44 and p47 were also up-regulated in pluripotent embryonic stem cells (ESCs) from mice and human, respectively, where they maintained a highly proliferative and undifferentiated state through inhibition of the differentiation program driven by p53fl [90]. p47 expression was also found up-regulated in highly proliferating primary human neural progenitor cells in culture (Figure 2) [91]. Interestingly, the common glial/neuronal precursor cell (NSCs) was proposed as the cell-of-origin of glioblastoma multiforme (GBM) [92]. Altogether, these observations suggest that p47 might play a role in cancer development under specific conditions. In this regard, GBM primary samples, xenografts and gliosis (reactive astrocytosis) showed a consistently elevated expression of p47 while the levels of p53fl were low or absent [91]. This proposes that proliferating brain cells may express more p47, and that a higher p47:p53 ratio is a distinguishing feature of tumor vs. non-tumor in brain tissues [91]. Furthermore, p47 might also play a role in other forms of cancers since it was detected overexpressed in melanoma cell lines compared to normal melanocytes, and in ovarian cancer samples where it was associated with lower overall survival [93,94,95].

Altogether, these data not only support the idea that p47 increases the diversity of the p53 response, but they also suggest that p47 carries on specific activities that differ from those displayed by its full-length counterpart. Their differential role might be particularly important in the context of aging and cancer, and probably regulated in a tissue-specific manner.

## 5. p47 Activity During ER Stress 

### 5.1. Regulation of Cell Cycle

The balance between p53-mediated control of G1 and G2 phases of the cell cycle are both related to the control of p21 expression, although through different mechanisms (Table 1). During the DDR, the activation of p53fl induces p21 expression that allows to repair damaged DNA before entering to the replication phase [20]. The induction of p47, devoid of the TAI domain, prevented p53fl-mediated activation of the p21 promoter and, importantly, it also prevented the translation of *p21* mRNA (Table 1) [31,62,82]. In line with this observation, intraperitoneal injections of tunicamycin in mice resulted in ER stress, as evaluated by CHOP expression, followed by a decrease of p21 protein levels in various organs such as lungs, heart, liver, spleen, and pancreas [96]. This depletion, in turn, resulted in increased levels of 14-3-3σ, through inhibition of its COP1-mediated degradation [31]. In addition, p47, but not p53fl, directly induces the expression of 14-3-3σ at the transcriptional level [58,80]. 14-3-3σ sequesters the cyclin B1/CDC2 complex in the cytoplasm, thereby arresting the cells in the G2/M phase [97]. Therefore, p53-dependent control of p21 appears as a master regulator of cell cycle arrest as part of the cellular responses triggered by both DNA damage and ER stress (Table 1).

### 5.2. Induction of Apoptosis

Apart from the cell cycle regulatory role described before, p47 also plays a role in promoting apoptosis when the cell trauma is too severe. Down-regulation of 14-3-3σ by siRNA transfection was shown to magnify p47-mediated cell death during ER stress [31,58]. This supports the notion that a pro-apoptotic response is more likely if the cell fails to arrest in G2 and to restore the ER homeostasis, in a similar fashion as the p53fl-dependent apoptosis is triggered when cells fail to arrest in G1 following genotoxic stress [20,25]. Interestingly, induction of apoptosis by p53 during ER stress relies, at least in part, on the suppression of BiP synthesis (Table 1) [98]. In addition to its chaperoning activity, BiP binds several partners, including the inducers of the UPR axis, that together control cell growth and protect against stress-induced apoptosis [99,100,101,102,103,104,105,106]. Accordingly, BiP is upregulated in various cancers, including brain, prostate, head and neck and melanoma [41,107,108,109,110]. Though the molecular mechanisms remain largely unknown, BiP’s mRNA interaction with p53 resulted in inhibition of BiP protein synthesis and therefore less interaction between BiP and Bcl-2-interacting killer (BIK), a pro-apoptotic member of the Bcl-2 family [98,106]. Remarkably, the p53-dependent drop in BiP/BIK association only occurred during ER stress, implying that stress conditions are also key in controlling the BiP-BIK interaction and cell fate decisions [98]. This observation supports previous results showing that BiP expression was counteracted by p53 in MEFs, HCT116, and U2OS cell lines during brefeldin A and tunicamycin-induced ER stress [111]. According to the protective roles of BiP, inhibition of its synthesis by p53 contributes to the onset of apoptosis and therefore has a profound impact in cellular fate under ER stress [39,41,98]. What is not yet clear and under investigation is if the suppression of BiP sensitizes the cells to ER stress in general or if p53 has an additional effect that makes the BiP-BIK interaction a specific target. As p53 only affects the interaction under ER stress conditions when BiP is induced, it is possible that other factors contribute to the selection of which of BiP’s interactions are targeted.

### 5.3. Regulation of mRNA Translation

The tumor suppressor p53 regulates the expression of translation-related factors such as eIF4F, the ternary complex and ribosomal RNA and proteins, thereby fine-tuning protein synthesis and cell growth [112,113]. This leads to tumor-suppressive programs imposed at the translational level that were overlooked by studies assessing alterations in gene expression only through transcriptional changes, but that became central when the translatome was analyzed [113]. Indeed, activation of p53 by doxorubicin or nutlin-3a in MCF-7 cells revealed a considerable uncoupling between transcription and translation, suggesting that post-transcriptional control of gene expression is a major aspect of p53-regulated responses [113,114]. In cells with active UPR, general cap-dependent translation is much inhibited via PERK-dependent eIF2α phosphorylation to avoid protein load into the ER, while the cap-independent counterpart is facilitated to promote synthesis of specific proteins required to cope with the stress [75,76,77]. In this context, regulation of mRNA translation appears as an effective mechanism to target protein expression levels and to control cellular fate. Also, the selective induction of p47 translation during ER stress strongly supports this idea.

The robust translational control exerted by p53 during ER stress also contributes to the modulation of cell cycle and apoptosis [30,31,98]. Interestingly, some of these activities depend on p53’s RNA-binding capacity, a much less characterized property of p53 compared to the well-studied DNA-binding activity. Although p53 has been recognized as both a DNA- and RNA-binding protein (DRBP), a property shared by at least 2% of the human proteome [115], only a few RNAs have been found to directly interact with p53 [98,116,117,118,119,120,121,122,123]. The RNA binding capacity of p53 was proposed to be located on its DB domain [122], a shared region between p53fl and p47. Although a short domain localized to the first seven aa of p47 was found to be required to suppress translation [98], the interacting partners involved are unknown and are under active investigation. Nevertheless, the lack of the p53 N-terminal region allows p47 to oligomerize more easily, a characteristic that could explain why a p47-dependent phenotype is observed in the presence of p53fl during ER stress [31,58,80]. It is possible that a coherent translational program orchestrates the expression of other cell cycle and apoptosis regulatory factors [113,114], in particular when facing stress to the ER. The identification of transcripts subjected to this regulation will help to better characterize this aspect of p53 activity.

## 6. Therapeutic Approaches Based on p53

The p53 pathway offers opportunities for the development of therapies directed towards UPR-addicted cancers. It was recently reported that the human triple-negative breast cancer (TNBC) cell line MDA-MB-231 resistant to thapsigargin and tunicamycin-induced apoptosis, relied on DB domain mutants of p53 (mutp53) to inhibit the expression of PERK and IRE1α, and consequently their pro-death-associated responses [124]. Also, the activity of ATF6 was shown to be promoted by mutp53 leading to an adaptive output with higher cell aggressiveness in Matrigel invasion assays [124]. While these observations support the idea of a direct role of p53 in the control of the three UPR sensors, the mechanisms involved and the role of p47 are scarcely known, and thus require closer attention [124,125]. However, they still emphasize the importance to develop therapeutic strategies aiming to restore the normal activity of p53 in cancer cells and/or to remove the GOF properties linked to mutp53, thus activating the ER stress-dependent death pathways. In this regard, targeting mutp53 by using the FDA-approved histone-deacetylase inhibitor Suberoylanilide Hydroxamic Acid (SAHA) that has been shown to reduce mutp53 protein levels and leads to preferential death of mutp53-expressing cells [126], sensitized cells to ER stress-induced death [124]. This effect was even higher when ATF6 cleavage (i.e., activation) was inhibited by Nelfinavir (NFV) treatment [124], emphasizing the promises hold in targeting p53 in cancers with an active UPR. Despite the above-described results were attributed to the GOF activities displayed by mutp53 [124], work in myeloid malignancies showed no evidence of GOF for missense mutations found in p53 in human cancers and rather pointed to the DNE effect as the cause of impaired p53 function [7,127]. In that regard, the modulation of p53 activity by p47 during stress in the ER could be seen as a GOF activity when it promotes cellular responses that are not induced by p53fl [63,81], and/or as a dominant-negative regulator of some of p53fl classical target genes [30,31,60,62]. Therefore, to better understand this phenomenon, focus must be made on the functional implications of higher p47 expression on known p53-controlled cellular responses.

The association between p53, UPR and cancer therapy is further exemplified by the effect of proteasome inhibitors (PIs) in promoting p53-mediated cell death. Hematological malignancies are successfully treated with PIs alone or in combination therapy. Moreover, PIs are currently being tested for the treatment of solid tumors such as lung, colon, pancreas, breast and head and neck cancer [44,45,128]. The efficacy of such treatments relies on the higher sensitivity of cancer cells to the cytotoxic effects of PIs, due to their increased proliferation and requirement for protein synthesis, and appears to be related to altered proliferation and apoptosis signaling pathways [44]. The pro-death effects of PIs were shown to act at least in part through accumulation and activation of p53. This, in turn, modulates the expression of some down-stream pro- and anti-apoptotic target genes such as BAX, PUMA and Survivin in prostate, clear cell renal cell and colon cancer, and in murine mammary and rat fibroblasts cell lines [129,130,131,132,133]. Moreover, inactivation of p53 in PI-treated cells partially counteracted apoptosis induction [129,131]. Interestingly, it has been also shown that PI-promoted stimulation of NOXA, a pro-apoptotic member of the Bcl-2 family, and induced apoptosis in a p53-independent fashion in melanoma and myeloma cell lines but not in normal melanocytes [134]. Although these observations shed some light, the molecular mechanisms at place during PI-mediated induction of apoptosis remain partial and thus, the identification of other key players is still an open task. Importantly, whether p47 has an impact on p53fl activity and on the final cellular output upon PI treatment remains unknown.

## 7. Conclusions and Perspectives 

The role of p53 during the UPR is far from being definitely established. Although p53 could be either activated or repressed in this scenario, the conditions (i.e., systems, inducers, settings, etc.) tested are diverse, making it difficult to obtain general conclusions. In addition, most studies have focused on the full-length protein without addressing the role of the ER stress-induced isoform p47. Indeed, the physiological roles, as well as the stress-responsive effects mediated by p47, are also marginally described. The consistent expression of p47 in cultured cells challenged with ER stress suggests that it plays an important part in the p53 response during the UPR. Certainly, some specific roles might be attributed to it, like the inhibition of p53fl-mediated induction of p21 and the induction of 14-3-3σ transcription, both activities contributing to G2 arrest [31,58]. On the other hand, its role in controlling BiP synthesis and its effect on apoptosis under ER stress seem to be a shared capacity with the full-length protein [98]. Nevertheless, higher ratios of p47 might affect the interactome of p53fl and might impose changes on its activity. Overall, the mechanisms that promote the specific induction of p47 and the consequent cellular responses induced by this protein under different conditions are partially known. However, these open questions constitute an attractive field of research that could also contribute to the development of new therapeutic strategies for cancer treatment.

Particular importance must be paid to the mechanisms used by p53 to regulate mRNA translation. This role emerges as an interesting target since it offers a new way for p53 to dictate cell fate during ER stress, as exemplified by the cell cycle arrest and apoptosis induced by suppression of p21 and BiP, respectively (Table 1) [31,58,98]. Importantly, translational control of specific mRNA targets also provides a physiological role for p53’s RNA binding activity [70,98], which merits more attention. In addition, it illustrates how cells can differentially modulate p53 activity in response to a defined cellular stress pathway in order to trigger a specific and suitable cell-biological outcome by inducing the expression of different isoforms with apparent distinctive functions (Figure 1 and Table 1).

Finally, the role of p53 in UPR/PI-induced cell death is only partially understood. Importantly, whether p47 has an impact on p53fl activity and on the final cellular output upon PI treatment remains unknown. Considering its capacity to alter p53fl activity (DNE) or to exert new functions on its own (GOF), a thorough characterization of the cellular responses promoted by p47 and their underlying molecular mechanisms is an urgent need. The results obtained not only will open up for the identification of drugs that alter its expression under specific conditions, but they will also shed light on major human physiological aspects including ageing and cancer development, and they will contribute to the development of novel therapeutic approaches.

## Figures and Tables

**Figure 1 cancers-12-00401-f001:**
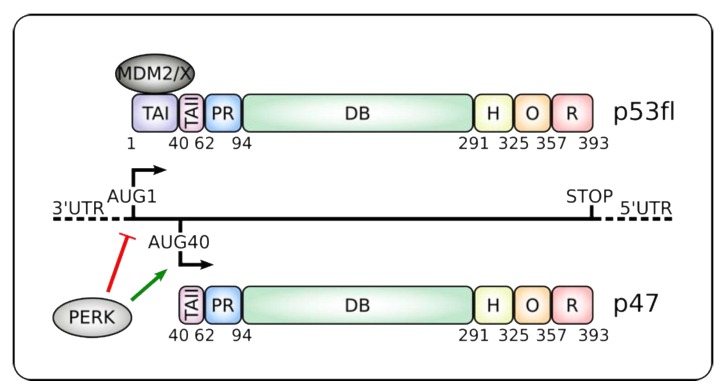
Alternative translation initiation products from the p53 mRNA. Alternative translation initiation from codons AUG1 or AUG40 gives rise to canonical p53 full-length (p53fl) or the shorter p47 isoforms, respectively. Endoplasmic reticulum (ER) stress and induction of the Unfolded Protein Response (UPR) leads to protein kinase RNA-like ER kinase (PERK) kinase activation that suppresses cap-dependent synthesis of p53fl and favors IRES-dependent translation initiation of p47 [58,59,60,61,62]. p47 lacks the MDM2 and MDMX binding sites and the transactivation domain I (TAI), which controls transcription of some typical p53-target genes such as p21 and MDM2 (see text and Table 1) [63]. p47 retains the oligomerization (O) domain and can form hetero- or homo-oligomers. It also keeps the Proline-rich (PR), DNA-binding (DB), hinge (H) and C-terminal regulatory (R) domains. Numbers below the proteins denote the location of the different domains. 3’ and 5’ UTRs: Untranslated regions.

**Figure 2 cancers-12-00401-f002:**
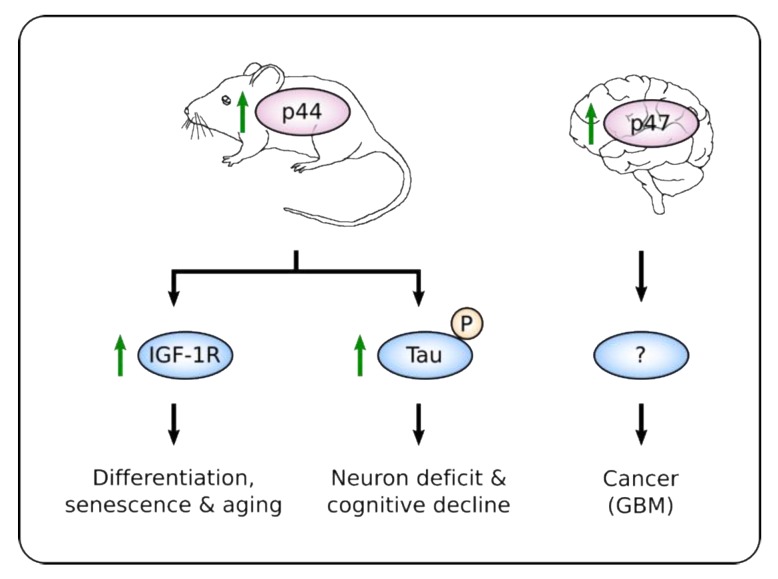
Cellular and physiological responses controlled by higher relative amount of p47/p44. Increased expression of p47/p44 in different tissues and circumstances leads to diverse outputs. High expression of p44 (mouse homolog of p47) promotes differentiation of embryonic stem cells (ESCs) [90] and senescence of several mice tissues leading to a premature aging phenotype through hyper-activation of insulin-like growth factor receptor (IGF-1R) [84]. In normal mice brain tissue, p44-induced expression of Tau kinases results in hyper-phosphorylation of Tau and neuron deficit and cognitive decline [87]. Although elevated expression of p47 was also reported in neural progenitors and in glioblastoma (GBM) samples, the signaling pathway involved is yet unknown [91].

**Table 1 cancers-12-00401-t001:** Examples of the different regulatory outputs mediated by p53 isoforms during the DNA damage response (DDR) and UPR.

Condition	Target	Mechanisms	Expression	Cellular Output	References
DDR	p21	Transcription	Induced	G1 arrest	[20,25]
	MDM2	Transcription	Induced	p53 regulation	[16,26,27,28]
	BAX	Transcription	Induced	Apoptosis	[23]
UPR	p21	Transcription & translation	Repressed	G2 arrest	[31,58]
	14-3-3σ	Transcription	Induced	G2 arrest	[58]
	MDM2	Translation	Repressed	?	[30]
	BiP	Translation	Repressed	Apoptosis	[98]

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
