# Peer review of "Alternative Mechanisms of p53 Action During the Unfolded Protein Response"

_cancers, 2020, doi:10.3390/cancers12020401_

Round 1

Reviewer 1 Report

The review by Fusee et al. is focussed on the role of p53 and particularly on the p53 isoform p47 in the Unfolded Protein Response (UPR) upon Endoplasmic Reticulum (ER) stress. The authors describe the literature on how the p53 pathway acts during the UPR to induce cell cycle arrest and/or apoptosis and put emphasis on the emerging role of p47 as a critical regulator of this process. Finally, the authors provide prospects on how compounds that alter the expression of p47 could be considered as interesting therapeutic approaches.

The review is well-written, focussed and provides an accurate description on our current understanding on the role of p53 on UPP and the ER-stress response. I have a few comments that the authors may find useful to address:

1.     The role of the proteasome inhibitors in the ER-stress and UPR should be described in more detail. Particularly, because the clinical efficacy of these inhibitors is mainly based on the UPR. The paragraph between lines 119-124 on this subject is rather vague. This part will be relevant with the Conclusions and perspectives paragraph.

2.     The authors describe the apparent controversies in the literature on the p53 response upon ER-stress and suggest that the emerging data on p47 may provide an explanation for this. However, at least to the reviewer, is not very clear the impact of p47 function on this aspect.

There are a few spelling mistakes:

Line 154: established instead of stablished

Line 263: tumors instead of tumorous

Author Response

Response to Reviewer 1 Comments.

Point 1. The role of the proteasome inhibitors in the ER-stress and UPR should be described in more detail. Particularly, because the clinical efficacy of these inhibitors is mainly based on the UPR. The paragraph between lines 119-124 on this subject is rather vague. This part will be relevant with the Conclusions and perspectives paragraph.

Response 1: This is an important subject and we have described the role of proteasome inhibitors in more detail. The paragraph between previous lines 119-124 is improved (please see new paragraph between lines 116-122). In addition, a new entire paragraph (lines 401-418) within the new section “Therapeutic approaches based on p53” was included and is exclusively dedicated to the roles of proteasome inhibitors in the UPR. This is also relevant to Reviewer 2 Comments (Point 2 “Conclusions and Perspectives”).

Point 2. The authors describe the apparent controversies in the literature on the p53 response upon ER-stress and suggest that the emerging data on p47 may provide an explanation for this. However, at least to the reviewer, is not very clear the impact of p47 function on this aspect.

Response 2: This is central to our MS and we thank the Reviewer for this comment. The manuscript aims to present the controversies found in the literature about p53 expression during the UPR and to highlight that some, at least, of the previous studies have not considered the p47 isoform. But with accumulating data from in cellulo and animal models, it is clear that this isoform plays a significant role. At this stage, only the UPR has been shown to be a physiological regulator of p47 expression and we therefore hope to draw further attention to this isoform as a potential key player in p53-mediated regulation of the UPR.

Point 3. Spelling mistakes

Response 3: The pointed mistakes, as well as others found throughout the text, were corrected.

Reviewer 2 Report

Fusee L et. al. discuss the activity of p53 in ER stress, in particular the differential role of transcript variants created by alternative translation start codons.  The authors cover several areas of interest (induction of splice variants, structure/function of variants, potentiation of cancer).  It is evident that the authors have researched the topic thoroughly however the manuscript would benefit from improvement.

There are 2 broad issues with the manuscript.

1 Language

Although the meaning behind the text is evident, the expression needs improvement as it contains oddly phrased regions of text.  The very first sentence of the Abstract is an example -

"The tumor suppressor protein p53 orchestrates cellular responses to cope with a vast number of stresses, being DNA damage and oncogenic activation some of the best described" could better be expressed as "The tumor suppressor protein p53 orchestrates cellular responses to respond to a vast number of stresses, with DNA damage and oncogenic activation being some of the best described". 

The manuscript would benefit from a thorough editing to improve the grammar, syntax and general readability.

2 Narrative

From section 3 (p53 expression and activity during ER stress) onwards the momentum of the manuscript dissipates and there appears to be less logical progression of the text, to the point where the order of paragraphs is interchangeable.  This is not necessarily a critical failing; however the authors may more successfully convey their perspective by the use of subheadings, better integration with figures and better highlighting the relevance to disease.  Additionally, the Conclusions and Perspectives section introduces new points and doesn’t integrate or distil previous sections.  It may help if the authors added a figure which, for example, illustrates the links between disease processes, ER stress, UPR and the physiological sequelae of different splice variants.

Author Response

Response to Reviewer 2 Comments.

Point 1. Language

Although the meaning behind the text is evident, the expression needs improvement as it contains oddly phrased regions of text. The very first sentence of the Abstract is an example - "The tumor suppressor protein p53 orchestrates cellular responses to cope with a vast number of stresses, being DNA damage and oncogenic activation some of the best described" could better be expressed as "The tumor suppressor protein p53 orchestrates cellular responses to respond to a vast number of stresses, with DNA damage and oncogenic activation being some of the best described". 

The manuscript would benefit from a thorough editing to improve the grammar, syntax and general readability.

Response 1: The general readability, syntax and grammar were improved and were also checked by external English speakers. In particular, the sentence mentioned by the Reviewer was corrected as suggested. Other oddly phrases were also corrected. This is also relevant to Reviewer 3 comments (Point 3 “Minor text editing”).

Point 2. Narrative

From section 3 (p53 expression and activity during ER stress) onwards the momentum of the manuscript dissipates and there appears to be less logical progression of the text, to the point where the order of paragraphs is interchangeable. This is not necessarily a critical failing; however the authors may more successfully convey their perspective by the use of subheadings, better integration with figures and better highlighting the relevance to disease.  Additionally, the Conclusions and Perspectives section introduces new points and doesn’t integrate or distil previous sections.  It may help if the authors added a figure which, for example, illustrates the links between disease processes, ER stress, UPR and the physiological sequelae of different splice variants.

Response 2: We thank the Reviewer for this helpful comment. The logical progression of the text has been improved by relocating several paragraphs into a more “natural” position and by the use of subheadings in sections 3, 4 and 5 (see new text). The improved figures and the table created were better integrated according to the reassembled text.

The previous Conclusions and Perspectives section was transformed into the new section “Therapeutic approaches based on p53”. The actual concluding section does not introduce new aspects and rather summarizes and integrates the data presented throughout the MS as suggested by the Reviewer.

We have modified Figure 2 to focus on the known physiological and pathological roles of higher p47 expression in mice and humans. In addition, we have included a new table (Table 1) that summarizes the link among the higher levels of p53fl or p47 observed during the DDR and UPR, respectively, their effect on some classical and relevant p53-target genes, the molecular mechanism involved and the associated cellular output. This is also relevant to Reviewer 3 Comments (point 1 “p47 activity during ER stress”). We hope this will address the Reviewer’s concerns.

Reviewer 3 Report

The manuscript "Alternative mechanisms of p53 action during the Unfolded Protein Response " reviews the role of p53 during the Unfolded Protein Response (UPR), with a particular emphasis on
p53 isoforms that modulate the cellular response to p53 activation.

After an introduction into general p53 function and its loss during cancer progression, the authors describe how endoplasmic reticulum stress triggers the UPR, and how p53 is
activated during ER stress. The characteristics of p53 activation during ER stress differ from "canonical" activation of p53, which can be explained by the ER stress-specific induction of a shorter isoform
of p53, p47, which lacks one of the transactivation domains and the MDM2 binding site. The authors then summarize findings about p47 activity during ER stress, notably p53´s RNA binding properties, and how
this may be translated into treatment of UPR-addicted cancers.

This is an interesting review that summarizes the current knowledge about selective p53 isoform expression after ER stress.

Since the section "p47 activity during ER stress" is the most important section with regard to the general aim of the review, it would be helpful to expand current Figure 2, or have an additional figure illustrating the concepts
of p53/p47 function during e.g. translation inhibition of selected mRNAs (BIP). This particular aspect is not illustrated clearly by Figure 1.

The title of Figure 1 "Expression of translational variants from the p53 mRNA" should also include something like "and differences in regulatory output".
For clarity, it could also be split into two figures : One illustrating the generation of the relevant isoforms, and another showing the regulatory output, with the second figure being used
to illustrate the concepts of one of the later sections (e.g. "p47 activity during ER stress" ).

Minor text editing is necessary before being acceptable for publication. For example :
line 282 : "Secondly, it favors the translation of a cap-independent selected number of mRNAs, such as ATF ..." should be
"Secondly, it favors the cap-independent translation of a selected number of mRNAs, such as ATF ..." ?

Author Response

Response to Reviewer 3 Comments.

Point 1. Since the section "p47 activity during ER stress" is the most important section with regard to the general aim of the review, it would be helpful to expand current Figure 2, or have an additional figure illustrating the concepts of p53/p47 function during e.g. translation inhibition of selected mRNAs (BIP). This particular aspect is not illustrated clearly by Figure 1.

Response 1: This is right, this section is the most important and in order to better describe the activity of p47 the previous Figure 2 was modified so that now focuses on the known physiological and pathological roles of higher p47 expression in mice and humans. In addition, a table was created (Table 1) that summarizes the link among the higher levels of p53fl or p47 observed during the DDR and UPR, respectively, their effect on some classical and relevant p53-target genes, the molecular mechanism involved (i.e transcription and/or translation) and the associated cellular output. This is also relevant to Reviewer 2 Comments (point 2 “Additional Figure”).

Point 2. The title of Figure 1 "Expression of translational variants from the p53 mRNA" should also include something like "and differences in regulatory output".  For clarity, it could also be split into two figures : One illustrating the generation of the relevant isoforms, and another showing the regulatory output, with the second figure being used to illustrate the concepts of one of the later sections (e.g. "p47 activity during ER stress" ).

Response 2: The information included in previous Figure 1 was split into new Figure 1 and new Table 1. New Figure 1 only presents information about p53 isoforms and their origin in order to be better integrated with the text. The information presented in Table 1 is described above. This table integrates the concepts suggested by the Reviewer to be included in the “second new figure”. See also Response 2 to Reviewer 2.

Point 3. Minor text editing is necessary before being acceptable for publication. For example :
line 282 : "Secondly, it favors the translation of a cap-independent selected number of mRNAs, such as ATF ..." should be "Secondly, it favors the cap-independent translation of a selected number of mRNAs, such as ATF ..." ?

Response 3: The sentence mentioned by the Reviewer was corrected as suggested. Other oddly phrases were also corrected. This is also relevant to Reviewer 2 comments (Point 1 “Language”).

Round 2

Reviewer 2 Report

The review is much easier to navigate and read.

Some minor suggestions -

Ln 106
"This phenomenon is thought to explain the anti-cancer effects of proteasome inhibitors (PIs) that induce accumulation and aggregation of proteins that is less tolerated by highly proliferative and protein synthesis-demanding malignant cells (see below) [44,45]."

Could be -

"This phenomenon is thought to explain the anti-cancer effects of proteasome inhibitors (PIs) that induce accumulation and aggregation of proteins that ARE less tolerated by THE highly proliferative and protein synthesis-demanding malignant cells (see below) [44,45]."

or alternatively

"This phenomenon is thought to explain the anti-cancer effects of proteasome inhibitors (PIs) that induce THE accumulation and aggregation of proteins that is less tolerated by highly proliferative and protein synthesis-demanding malignant cells (see below) [44,45]."

Ln 219
... in cellulo...
"in vitro"

Author Response

Response to Reviewer 2 Comments.

Point 1. The Reviewer indicated that the following phrase required correction: "This phenomenon is thought to explain the anti-cancer effects of proteasome inhibitors (PIs) that induce accumulation and aggregation of proteins that is less tolerated by highly proliferative and protein synthesis-demanding malignant cells (see below) [44,45]."

Response 1: The phrase was corrected following Reviewer’s suggestion: "This phenomenon is thought to explain the anti-cancer effects of proteasome inhibitors (PIs) that induce accumulation and aggregation of proteins that are less tolerated by the highly proliferative and protein synthesis-demanding malignant cells (see below) [44,45]."

Point 2. The Reviewer indicated that the following phrase required correction: “The functional properties of p47 have been addressed in cellulo and in animal models by several groups.”

Response 2: The phrase was corrected following Reviewer’s suggestion: “The functional properties of p47 have been addressed in vitro and in animal models by several groups.”